# Improved Learning-augmented Algorithms for k-means and k-medians Clustering

**Thy Nguyen** [*], **Anamay Chaturvedi** [*], **Huy Lê Nguyễn** [*]
Khoury College of Computer Sciences,
Northeastern University
{nguyen.thy2,chaturvedi.a,hu.nguyen}@northeastern.edu

## Abstract

We consider the problem of clustering in the learning-augmented setting. We are given a data set in $d$-dimensional Euclidean space, and a label for each data point given by a predictor indicating what subsets of points should be clustered together. This setting captures situations where we have access to some auxiliary information about the data set relevant for our clustering objective, for instance the labels output by a neural network. Following prior work, we assume that there are at most an $\alpha \in (0, c)$ for some $c < 1$ fraction of false positives and false negatives in each predicted cluster, in the absence of which the labels would attain the optimal clustering cost OPT. For a dataset of size $m$, we propose a deterministic $k$-means algorithm that produces centers with an improved bound on the clustering cost compared to the previous randomized state-of-the-art algorithm while preserving the $O(dm \log m)$ runtime. Furthermore, our algorithm works even when the predictions are not very accurate, i.e., our cost bound holds for $\alpha$ up to $1/2$, an improvement from $\alpha$ being at most $1/7$ in previous work. For the $k$-medians problem we again improve upon prior work by achieving a biquadratic improvement in the dependence of the approximation factor on the accuracy parameter $\alpha$ to get a cost of $(1 + O(\alpha))$OPT, while requiring essentially just $O(md \log^3 m/\alpha)$ runtime.

## 1 Introduction

In this paper we study $k$-means and $k$-medians clustering in the learning-augmented setting. In both these problems we are given an input data set $P$ of $m$ points in $d$-dimensional Euclidean space and an associated distance function dist$(\cdot, \cdot)$. The goal is to compute a set $C$ of $k$ points in that same space that minimize the following cost function:

$$\text{cost}(P, C) = \sum_{p \in P} \min_{i \in [k]} \text{dist}(p, c_i).$$

In words, the cost associated with a singular data point is its distance to the closest point in $C$, and the cost of the whole data set is the sum of the costs of its individual points.

In the $k$-means setting dist$(x, y) := \|x - y\|^2$, i.e., the square of the Euclidean distance, and in the $k$-medians setting we set dist$(x, y) := \|x - y\|$, although here instead of the norm of $x - y$, we can in principle also use any other distance function. These problem are well-studied in the literature of algorithms and machine learning, and are known to be hard to solve exactly (Dasgupta, 2008), or even approximate well beyond a certain factor (Cohen-Addad & Karthik C. S., 2019). Although approximation algorithms are known to exist for this problem and are used widely in practice, the theoretical approximation factors of practical algorithms can be quite large, e.g., the 50-approximation in Song & Rajasekaran (2010) and the $O(\ln k)$-approximation in Arthur & Vassilvitskii (2006). Meanwhile, the algorithms with relatively tight approximation factors do not necessarily scale well in practice (Ahmadian et al., 2019).

To overcome these computational barriers, Ergun et al. (2022) proposed a *learning-augmented* setting where we have access to some auxiliary information about the input data set. This is motivated

---

[*]Equal contribution. All three authors were supported in part by NSF CAREER grant CCF-1750716 and NSF grant CCF-1909314.

by the fact that in practice we expect the dataset of interest to have exploitable structures relevant to the optimal clustering. For instance, a classifier's predictions of points in a dataset can help group similar instances together. This notion was formalized in Ergun et al. (2022) by assuming that we have access to a predictor in the form of a labelling $P = P_1 \cup \cdots \cup P_k$ (all the points in $P_i$ have the same label $i \in [k]$), such that there exist an unknown optimal clustering $P = P_1^* \cup \cdots \cup P_k^*$, an associated set of centers $C = (c_1^*, \ldots, c_k^*)$ that achieve the optimally low clustering cost OPT ( $\sum_{i \in [k]} \text{cost}(P_i, \{c_i^*\}) = \text{OPT}$), and a known *label error rate* $\alpha$ such that:

$$|P_i \cap P_i^*| \geq (1 - \alpha) \max(|P_i|, |P_i^*|)$$

In simpler terms, the auxiliary partitioning $(P_1, \ldots, P_k)$ is close to some optimal clustering: each predicted cluster has at most an $\alpha$-fraction of points from outside its corresponding optimal cluster, and there are at most an $\alpha$-fraction of points in the corresponding optimal cluster not included in predicted cluster. The predictor, in other words, has at most $\alpha$ false positive and false negative rate for each label.

Observe that even when the predicted clusters $P_i$ are close to a set of true clusters $P_i^*$ in the sense that the label error rate $\alpha$ is very small, computing the means or medians of $P_i$ can lead to arbitrarily bad solutions. It is known that for $k$-means the point that is allocated for an optimal cluster should simply be the average of all points in that cluster (this can be seen by simply differentiating the convex 1-mean objective and solving for the minimizer). However, a single false positive located far from the cluster can move this allocated point arbitrarily far from the true points in the cluster and drive the cost up arbitrarily high. This problem requires the clustering algorithms to process the predicted clusters in a way so as to preclude this possibility.

Using tools from the robust statistics literature, the authors of Ergun et al. (2022) proposed a randomized algorithm that achieves a $(1 + 20\alpha)$-approximation given a label error rate $\alpha < 1/7$ and a guarantee that each predicted cluster has $\Omega\left(\frac{k}{\alpha}\right)$ points. For the $k$-medians problem, the authors of Ergun et al. (2022) also proposed an algorithm that achieves a $(1 + \alpha')$-approximation if each predicted cluster contains $\Omega\left(\frac{n}{k}\right)$ points and a label rate $\alpha$ at most $O\left(\frac{\alpha'^4}{k \log \frac{k}{\alpha'}}\right)$, where the big-Oh notation hides some small unspecified constant, and $\alpha' < 1$.

The restrictions for the label error rate $\alpha$ to be small in both of the algorithms of Ergun et al. (2022) lead us to investigate the following question:

*Is it possible to design a $k$-means and a $k$-medians algorithm that achieve $(1 + \alpha)$-approximate clustering when the predictor is not very accurate?*

## 1.1 OUR CONTRIBUTIONS

In this work, we not only give an affirmative answer to the question above for both the $k$-means and the $k$-medians problems, our algorithms also have improved bounds on the clustering cost, while preserving the time complexity of the previous approaches and removing the requirement on a lower bound on the size of each predicted cluster.

For learning-augmented $k$-means, we modify the main subroutine of the previous randomized algorithm to get a deterministic method that works for all $\alpha < 1/2$, which is the natural breaking point (as explained below). In the regime where the $k$-means algorithm of Ergun et al. (2022) applies, we get improve the approximation factor to $1 + 7.7\alpha$. For the larger domain $\alpha \in [0, 1/2)$, we derive a more general expression as reproduced in table 1. Furthermore, our algorithm has better bound on the clustering cost compared to that of the previous approach, while preserving the $O(md \log m)$ runtime and not requiring a lower bound on the size of each predicted cluster.

Our $k$-medians algorithm improves upon the algorithm in Ergun et al. (2022) by achieving a $(1 + O(\alpha))$-approximation for $\alpha < 1/2$, thereby improving both the range of $\alpha$ as well as the dependence of the approximation factor on the label error rate from bi-quadratic to near-linear. For success probability $1 - \delta$, our runtime is $O(\frac{1}{1-2\alpha} md \log^3 \frac{m}{\alpha} \log \frac{k \log(k/\delta)}{(1-2\alpha)\delta} \log \frac{k}{\delta})$, so we see that by setting $\delta = 1/\text{poly}(k)$, we have just a logarithmic dependence in the run-time on $k$, as opposed to a polynomial dependence.

| Work, Problem | Approx. Factor | Label Error Range | Time Complexity |
|---|---|---|---|
| Ergun et al. (2022), $k$-Means | $1 + 20\alpha$ | $(\frac{10\log m}{\sqrt{m}}, 1/7)$ | $O(md\log m)$ |
| Algorithm 1, $k$-Means | $1 + \frac{5\alpha - 2\alpha^2}{(1-2\alpha)(1-\alpha)}$ $1 + 7.7\alpha$ | $[0,1/2)$ $[0, 1/7)$ | $O(md\log m)$ |
| Ergun et al. (2022), $k$-Medians | $1 + \tilde{O}((k\alpha)^{1/4})$ | small constant | $O(md\log^3 m +$ $\text{poly}(k, \log m))$ |
| Algorithm 2, $k$-Medians | $1 + \frac{7+10\alpha-10\alpha^2}{(1-\alpha)(1-2\alpha)}$ | $[0, 1/2)$ | $\tilde{O}\left(\frac{1}{1-2\alpha}md\right.$ $\left.\log^3 m \log^2 \frac{k}{\delta}\right)$ |

Table 1: Comparison of learning-augmented $k$-means and $k$-medians algorithms. We recall that $m$ is the data set size, $d$ is the ambient dimension, $\alpha$ is the label error rate, and $\delta$ is the failure probability (where applicable). The success probability of the $k$-medians algorithm of Ergun et al. (2022) is $1 - \text{poly}(1/k)$. The $\tilde{O}$ notation hides some log factors to simplify the expressions.

**Upper bound on** $\alpha$. Note that if the error label rate $\alpha$ equals $1/2$, then even for three clusters there is no longer a clear relationship between the predicted clusters and the related optimal clusters - for instance given three optimal clusters $P_1^*, P_2^*, P_3^*$ with equally many points, if for all $i \in [3]$, the predicted clusters $P_i$ consist of half the points in $P_i^*$ and half the points in $P_{(i+1) \bmod 3}^*$, then the label error rate $\alpha = 1/2$ is achieved, but there is no clear relationship between $P_i^*$ and $P_i$. In other words, it is not clear whether the predicted labels give us any useful information about an optimal clustering. In this sense, $\alpha = 1/2$ is in a way a natural stopping point for this problem.

## 1.2 RELATED WORK

This work belongs to a growing literature on learning-augmented algorithms. Machine learning has been used to improve algorithms for a number of classical problems, including data structures (Kraska et al., 2018; Mitzenmacher, 2018; Lin et al., 2022), online algorithms (Purohit et al., 2018), graph algorithms (Khalil et al., 2017; Chen et al., 2022a;b), computing frequency estimation (Du et al., 2021) , caching (Rohatgi, 2020; Wei, 2020), and support estimation (Eden et al., 2021). We refer the reader to Mitzenmacher & Vassilvitskii (2020) for an overview and applications of the framework.

Another relevant line of work is clustering with side information. The works Balcan & Blum (2008); Awasthi et al. (2014); Vikram & Dasgupta (2016) studied an interactive clustering setting where an oracle interactively provides advice about whether or not to merge two clusters. Basu et al. (2004) proposed an active learning framework for clustering, where the algorithm has access to a predictor that determines if two points should or should not belong to the same cluster. Ashtiani et al. (2016) introduced a semi-supervised active clustering framework where the algorithm has access to a predictor that answers queries whether two particular points belong in an optimal clustering. The goal is to produce a $(1 + \alpha)$-approximate clustering while minimizing the query complexity to the oracle.

Approximation stability, proposed in Balcan et al. (2013), is another assumption proposed to circumvent the NP-hardness of approximation for $k$-means clustering. More formally, the concept of $(c, \alpha)$-stability requires that every $c$-approximate clustering is $\alpha$-close to the optimal solution in terms of the fraction of incorrectly clustered points. This is different from our setting, where at most an $\alpha$ fraction of the points are incorrectly clustered and can worsen the clustering cost arbitrarily.

Gamlath et al. (2022) studies the problem of $k$-means clustering in the presence of noisy labels, where the cluster label of each point created by either an adversarial or a random perturbation of the optimal solution. Their Balanced Adversarial Noise Model assumes that the size of the symmetric difference between the predicted cluster $P_i$ and optimal cluster $P_i^*$ is bounded by $\alpha|P_i^*|$. The algorithm uses a subroutine with runtime exponential in $k$ and $d$ for a fixed $\alpha \leq 1/4$. In this work, we have different assumptions on the predicted cluster cluster $P_i$ and the optimal cluster $P_i^*$. Moreover, our focus is on efficient algorithms practical nearly linear-time algorithms that can scale to very large datasets for $k$-means and $k$-medians clustering.

## 2 $k$-MEANS

---

**Algorithm 1** Deterministic Learning-augmented $k$-Means Clustering

---

**Require:** Data set $P$ of $m$ points, Partition $P = P_1 \cup \ldots P_k$ from a predictor, accuracy parameter $\alpha$

    **for** $i \in [k]$ **do**

        **for** $j \in [d]$ **do**

            Let $\omega_{i,j}$ be the collection of all subsets of $(1 - \alpha)m_i$ contiguous points in $P_{i,j}$.

            $I_{i,j} \leftarrow \operatorname{argmin}_{Z \in \omega_{i,j}} \operatorname{cost}(Z, \overline{Z}) = \operatorname{argmin}_{Z \in \omega_{i,j}} \sum_{z \in Z} z^2 - \frac{1}{|Z|} \left( \sum_{z' in Z} z' \right)^2$

        **end for**

        Let $\widehat{c}_i = (\overline{I_{i,j}})_{j \in [d]}$

    **end for**

    Return $\{\widehat{c}_1, \ldots, \widehat{c}_k\}$

---

We briefly recall some notation for ease of reference.

**Definition 1.** *We make the following definitions:*

1. *The given data set is denoted as $P$, and $m := |P|$. The output of the predictor is a partition $(P_1, \ldots P_k)$ of $P$. Further, $m_i := |P_i|$.*

2. *There exists an optimal partition $(P_1^*, \ldots, P_k^*)$ and centers $(c_1^*, \ldots, c_k^*)$ such that $\sum_{i \in [k]} \operatorname{cost}(P_i^*, c_i^*) = \text{OPT}$, the optimally low clustering cost for the data set $P$. Further-more, $m_i^* := |P_i^*|$. For each cluster $i \in [k]$, denote the set of true positives $P_i^* \cap P_i = Q_i$. Recall that $|Q_i| \geq (1 - \alpha) \max(|P_i|, |P_i^*|)$, for some $\alpha < 1/2$.*

3. *We denote the average of a set $X$ by $\overline{X}$. For the sets $X_i$ and $P_i$ we denote their projections onto the $j$-th dimension by $X_{i,j}$ and $P_{i,j}$, respectively.*

Before we describe our algorithm, we recall why the naive solution of simply taking the average of each cluster provided by the predictor is insufficient. Consider $P_i$, the set of points labeled $i$ by the predictor. Recall that the optimal 1-means solution for this set is its mean, $\overline{P_i}$. Since the predictor is not perfect, there might exist a number of points in $P_i$ that are not actually in $P_i^*$. Thus, if the points in $P_i \setminus P_i^*$ are significantly far away from $\overline{P_i^*}$, they will increase the clustering cost arbitrary if we simply use $\overline{P_i}$ as the center. The following well-known identity formalizes this observation.

**Lemma 2** (Inaba et al. (1994)). *Consider a set $X \subset \mathbb{R}^d$ of size $n$ and $c \in \mathbb{R}^d$,*

$$\operatorname{cost}(X, c) = \min_{c' \in \mathbb{R}^d} \operatorname{cost}(X, c') + n \cdot \|c - \overline{X}\|^2 = \operatorname{cost}(X, \overline{X}) + n \cdot \|c - \overline{X}\|^2.$$

Ideally, we would like to be able to recover the set $Q_i = P_i \cap P_i^*$ and use the average of $Q_i$ as the center. We know that $|Q_i \setminus P_i^*| \leq \alpha m_i^*$. By lemma 3, it is not hard to show that $\operatorname{cost}(P_i^*, \overline{Q_i}) \leq \left(1 + \frac{\alpha}{1-\alpha}\right) \operatorname{cost}(P_i^*, \overline{P_i^*}) = \left(1 + \frac{\alpha}{1-\alpha}\right) \operatorname{cost}(P_i^*, c_i^*)$, which also implies a $\left(1 + \frac{\alpha}{1-\alpha}\right)$- approximation for the problem.

**Lemma 3.** *For any partition $J_1 \cup J_2$ of a set $J \subset \mathbb{R}$ of size $n$, if $|J_1| \geq (1 - \lambda)n$, then $|\overline{J} - \overline{J}_1|^2 \leq \frac{\lambda}{(1-\lambda)n} \operatorname{cost}(J, \overline{J})$.*

Since we do not have access to $Q_i$, the main technical challenge is to filter out the outlier points in $P_i$ and construct a center close to $\overline{Q_i}$. Minimizing the distance of the center to $\overline{Q_i}$ implies reducing the distance to $c_i^*$ as well as the clustering cost.

Our algorithm for $k$-means, algorithm 1, iterates over all clusters given by the predictor and finds a set of contiguous points of size $(1 - \alpha)m_i$ with the smallest clustering cost in each dimension. At the high level, our analysis shows that the average of the chosen points, $I_{i,j}$, is not too far away from that of the true positives, $Q_{i,j}$. This also implies that the additive clustering cost of $\overline{I_{i,j}}$ would not be too large. Since we can analyze the clustering cost by bounding the cost in every cluster $i$ and

dimension $j$, for simplicity we will not refer to a specific $i$ and $j$ when discussing the intuition of the algorithm. The proofs of the following lemmas and theorem are included in the appendix.

Note that there can be multiple optimal solutions in the optimization step. The algorithm can either be randomized by choosing an arbitrary set, or can also be deterministic by always choosing the first optimal solution. Lemma 4 shows that the optimization step guarantees that $I_{i,j}$ has the smallest clustering cost with respect to all sets of size $(1-\alpha)m_i$ in $P_{i,j}$.

**Lemma 4.** *For all $i \in [k], j \in [d]$, let $\omega'_{i,j}$ be the collection of all subsets of $(1-\alpha)m_i$ points in $P_{i,j}$. Then*

$$\text{cost}(I_{i,j}, \overline{I_{i,j}}) = \min_{Z' \in \omega'_{i,j}} \text{cost}(Z', \overline{Z'}).$$

Since we know that $|Q_{i,j}| \geq (1-\alpha)m_i$, it can be shown from lemma 4 that the cost of the set $I_{i,j}$ is smaller than that of $Q_{i,j}$. More precisely,

$$\text{cost}(I_{i,j}, \overline{I_{i,j}}) \leq \frac{(1-\alpha)m_i}{|Q_i|} \text{cost}(Q_{i,j}, \overline{Q_{i,j}}). \tag{1}$$

With this fact, we are ready to bound the clustering cost by bounding $|\overline{I_{i,j}} - \overline{Q_{i,j}}|^2$,

$$|\overline{I_{i,j}} - \overline{Q_{i,j}}|^2 \leq 2|\overline{I_{i,j}} - \overline{I_{i,j} \cap Q_{i,j}}|^2 + 2|\overline{I_{i,j} \cap Q_{i,j}} - \overline{Q_{i,j}}|^2.$$

Using lemma 3, we can bound $|\overline{I_{i,j}} - \overline{I_{i,j} \cap Q_{i,j}}|^2$ and $|\overline{I_{i,j} \cap Q_{i,j}} - \overline{Q_{i,j}}|^2$ respectively by $\text{cost}(I_{i,j}, \overline{I_{i,j}})$ and $\text{cost}(Q_{i,j}, \overline{Q_{i,j}})$. Combining this fact with eq. (1), we can bound, $|\overline{I_{i,j}} - \overline{Q_{i,j}}|^2$ by $\text{cost}(Q_{i,j}, \overline{Q_{i,j}})$.

**Lemma 5.** *The following bound holds:*

$$|\overline{I_{i,j}} - \overline{Q_{i,j}}|^2 \leq \frac{4\alpha}{1-2\alpha} \frac{\text{cost}(Q_{i,j}, \overline{Q_{i,j}})}{|Q_i|}.$$

Notice that lemma 5 also applies to any set in $\omega_{i,j}$ with cost smaller than the expected cost of a subset of size $(1-\alpha)m_i$ drawn uniformly at random from $Q_{i,j}$. Instead of repeatedly sampling different subsets of $Q_{i,j}$ and returning the one with the lowest clustering cost, the optimization step not only simplifies the analysis of the algorithm, but also guarantees that we find such a subset efficiently. This is the main innovation of the algorithm.

In the notations of lemma 2, we can consider $c = \overline{I_{i,j}}, P^*_{i,j} = X, m^*_i = n$. Thus, we want to bound $|\overline{P^*_{i,j}} - \overline{I_{i,j}}|^2$ by $\frac{\text{cost}(P^*_{i,j}, \overline{P^*_{i,j}})}{m^*_i}$ to achieve a $(1 + O(\alpha))$-approximation. Recall that we bound $|\overline{I_{i,j}} - \overline{Q_{i,j}}|^2$ by $\frac{\text{cost}(Q_{i,j}, \overline{Q_{i,j}})}{|Q_i|}$ in lemma 5. In lemma 6 we relate $\text{cost}(Q_{i,j}, \overline{Q_{i,j}})$ to $\text{cost}(P^*_{i,j}, \overline{P^*_{i,j}})$ as follows,

$$\text{cost}(P^*_{i,j}, \overline{P^*_{i,j}}) \geq \frac{1-\alpha}{\alpha} m^*_i |\overline{P^*_{i,j}} - \overline{Q_{i,j}}|^2 + \text{cost}(Q_{i,j}, \overline{Q_{i,j}})$$

We can then apply lemma 5 to bound $|\overline{P^*_{i,j}} - \overline{I_{i,j}}|^2$ by $\frac{\text{cost}(P^*_{i,j}, \overline{P^*_{i,j}})}{m^*_i}$.

**Lemma 6.** *The following bound holds:*

$$|\overline{P^*_{i,j}} - \overline{I_{i,j}}|^2 \leq \text{cost}(P^*_{i,j}, \overline{P^*_{i,j}}) \left( \frac{\alpha}{1-\alpha} + \frac{4\alpha}{(1-2\alpha)(1-\alpha)} \right) / m^*_i$$

Applying lemma 6 and lemma 2 to all $i \in [K], j \in [d]$, we are able to bound the total clustering cost.

**Theorem 7.** *Algorithm 1 is a deterministic algorithm for $k$-means clustering such that given a data set $P \in \mathbb{R}^{m \times d}$ and a partition $(P_1, \ldots, P_k)$ with error rate $\alpha < 1/2$, it outputs a $\left(1 + \left(\frac{\alpha}{1-\alpha} + \frac{4\alpha}{(1-2\alpha)(1-\alpha)}\right)\right)$-approximation in time $O(dm \log m)$.*

**Corollary 8.** *For $\alpha \leq 1/7$, algorithm 1 achieves a clustering cost of $(1 + 7.7\alpha)\text{OPT}$.*

## 3   $k$-MEDIANS

In this section we describe our algorithm for learning-augmented $k$-medians clustering and a theoretical bound on the clustering cost and the run-time. Our algorithm works for ambient spaces equipped with any metric $\text{dist}(\cdot, \cdot)$ for which it is possible to efficiently compute the geometric median, which is the minimizer of the 1-medians clustering cost. For instance, it is known from prior work (Cohen et al., 2016) that the geometric median with respect to the $\ell_2$-metric can be efficiently calculated, and appealing to this result as a subroutine allows us to derive a guarantee for learning-augmented $k$-medians with respect to the $\ell_2$ norm.

**Theorem 9.** *(Cohen et al. (2016)) There is an algorithm that computes a $(1 + \epsilon)$-approximation to the geometric median of a set of size $n$ in $d$-dimensional Euclidean space with respect to the $\ell_2$ distance metric with constant probability in $O(nd \log^3(n/\epsilon))$ time.*

Looking ahead at the pseudocode of algorithm 2, we see that to eventually derive a bound on the time complexity, we would need to account for adjusting the success probability in the many calls to theorem 9.

**Corollary 10.** *It follows from theorem 9 that with probability $1 - \frac{\delta}{2k}$, we have that for all $j \in [R]$, $\text{cost}(P_i \setminus P_i', \widehat{c}_i^j)$ is a $(1 + \gamma)$-approximation to the optimal 1-median cost for $P_i \setminus P_i'$ while taking time $O(m_i d \log^3(m_i/\gamma) \log(Rk/\delta))$.*

We refer the reader to definition 1 for all notation that is undefined in this section; the only additional notation we introduce is the following definition.

**Definition 11.** *We make the following definitions:*

1. *We denote the optimal clustering cost of $P$ by $\text{OPT}$, and the optimal 1-median clustering cost of $P_i^*$ by $\text{OPT}_i$, with which notation we have that $\sum_{i \in [k]} \text{OPT}_i = \text{OPT}$.*

2. *We denote the distance $\text{dist}(x, y)$ between two points by $\|x - y\|$.*

We now describe at a high-level a run of our algorithm. Algorithm 2 operates sequentially on each cluster estimate; for the cluster estimate $P_i$, it samples a point $x \in P_i$ uniformly at random, and removes from $P_i$ the $\lceil \alpha m_i \rceil$-many points that lie furthest from $x$. It then computes the median of the clipped set, which is where we appeal to an algorithm for the geometric median, for instance theorem 9 when the ambient metric for the input data set is the $\ell_2$ metric. It turns out that this subroutine already gives us a good median for the cluster $P_i^*$ with constant probability (lemma 14); to boost the success probability we repeat this subroutine some $R$-many times (the exact expression is given in the pseudocode and justified in lemma 15), and pick the median with the lowest cost, denoted $\widehat{c}_i$. Collecting the $\widehat{c}_i$ across $i \in [k]$, we get our final solution $\{\hat{c}_1, \ldots, \hat{c}_k\}$.

---

**Algorithm 2** Learning-augmented $k$-Medians Clustering

**Require:** Data set $P$ of $m$ points, Partition $P = P_1 \cup \ldots P_k$ from a predictor, accuracy parameter $\alpha < 1/2$
    **for** $i \in [k]$ **do**
        Let $R = \frac{2}{1-2\alpha} \log \frac{2k}{\delta}$
        **for** $j \in [R]$ **do**
            Sample $x \sim P_i$ u.a.r.
            Let $P_i'$ be the $\lceil \alpha m_i \rceil$ points farthest from $x$
            $\widehat{c}_i^j \leftarrow$ median of $P_i \setminus P_i'$.
        **end for**
        Let $\widehat{c}_i$ be the $\widehat{c}_i^j$ with minimum cost
    **end for**
    Return $\{\hat{c}_1, \ldots, \hat{c}_k\}$

---

Although our algorithm itself is relatively straightforward, the analysis turns out to be more involved. We trace the proof at a high level in this section and mention the main steps, and defer all proofs to the appendix.

We see that it would suffice to allocate a center that works well for the true cluster $P_i^*$, but we only have access to the set $P_i$ with the promise that they have a significant overlap (as characterized by $\alpha$). Fixing an arbitrary true median $c_i^*$, one key insight is that the "false" points, i.e. points in $P_i \setminus P_i^*$ will only significantly distort the median if they happen to lie far from $c_i^*$. If there were a way to identify and remove these false points which lie far from $c_i^*$, then simply computing the geometric median of the clipped data set should work well.

By a direct application of Markov's inequality it is possible to show that a point $x$ picked uniformly at random will in fact lie at a distance on the order of the average clustering cost $OPT_i/m_i$ with constant probability, as formalized in lemma 12.

**Lemma 12.** *With probability $\frac{1-2\alpha}{2}$, $\|x - c_i^*\| \leq 2\mathrm{OPT}_i/m_i$.*

As we will condition on this good event holding, it will be convenient to introduce the notation $\mathcal{E}$.

**Definition 13.** *We let $\mathcal{E}$ denote the event that $\|x - c_i^*\| \leq 2\mathrm{OPT}_i/m_i$.*

Having identified a good point $x$ to serve as a proxy for where the true median $c_i^*$ lies, we need to figure out a good way to clip the data set so as to avoid false points which lie very far from $c_i^*$. We observe that since there are guaranteed to be at most $\lceil \alpha n \rceil$-many false points, if we were to remove the $\lceil \alpha n \rceil$-many points that lie farthest from $x$ (denoted $P_i'$), then we either remove false points that lie very far from $c_i^*$, or true points ($P_i^* \cap P_i'$) which are at the same distance from $c_i^*$ as the remaining false points (the points in $P_i \setminus (P_i' \cup P_i^*)$). In particular, this implies that the impact of the remaining false points is roughly dominated by the clustering cost of an equal number of true points, and we are able to exploit this to show that the clustering cost of $P_i \setminus P_i'$ with respect to its own median estimate $\widehat{c}_i$ is already close to that of the true center $P_i^*$.

**Lemma 14.** *Conditioned on $\mathcal{E}$, $\mathrm{cost}(P_i \setminus P_i', \widehat{c}_i^j) \leq (1 + 5\alpha)\mathrm{OPT}_i$.*

Since the event $\mathcal{E}$ that the randomly sampled point $x$ is close to a true median $c_i^*$ is true only with constant probability, we boost the success probability by running this subroutine some $R$ times and letting $\widehat{c}_i$ be the median estimate with respect to which the respective clipped data set had the lowest clustering cost.

**Lemma 15.** *For $R = O\left(\frac{1}{(1-2\alpha)} \log\left(\frac{2k}{\delta}\right)\right)$ many repetitions, with probability at least $1 - \frac{\delta}{2k}$, we have that $\mathrm{cost}(P_i \setminus P_i', \widehat{c}_i) \leq (1 + 5\alpha)\mathrm{OPT}_i$.*

We see from lemma 21 that the set $P_i \setminus P_i'$ differs from the true positives $P_i \cap P_i^*$ by sets of size at most $\lceil \alpha n \rceil$. It follows that as long as the distance between $\widehat{c}_i$ and $c_i^*$ is on the order of $\mathrm{OPT}_i/n$, they will not influence the clustering cost by more than an $O(\alpha\mathrm{OPT}_i)$ additive term, which we will be able to absorb into the $(1 + O(\alpha))$ multiplicative approximation factor. We formalize this in lemma 16.

**Lemma 16.** *If $\mathrm{cost}(P_i \setminus P_i', \widehat{c}_i) \leq (1 + 5\alpha)\mathrm{OPT}$, then $\|\widehat{c}_i - c_i^*\| \leq \frac{2+5\alpha}{(1-2\alpha)} \frac{\mathrm{OPT}_i}{n}$.*

We finally put everything together to show that the clustering cost of the set of true points $P_i \cap P_i^*$ with respect to the estimate $\widehat{c}_i$ is only at most an additive $O(\alpha\mathrm{OPT}_i)$ more than the cost with respect to the true median $c_i^*$. The key technical point in the analysis is that we can only appeal to the fact that the cost of $P_i \setminus P_i'$ is close to $OPT_i$, and we cannot directly reason about $\widehat{c}_i$ apart from appealing to lemma 16.

**Lemma 17.** *With probability $1 - \delta/k$, $\mathrm{cost}(P_i \cap P_i^*, \widehat{c}_i) \leq \mathrm{cost}(P_i \cap P_i^*, c_i^*) + \frac{(5\alpha+10\alpha^2)\mathrm{OPT}_i}{1-2\alpha}$.*

We can now derive our main cost bound stated in lemma 18. Doing so only requires that we account for the mislabelled points $P_i^* \setminus P_i$ which were not accounted for during our clustering. Again, from lemma 16 it suffices to appeal to the fact that the estimate $\widehat{c}_i$ lies within an $O(\alpha\mathrm{OPT}_i/n)$ distance of the true median $c_i^*$.

**Lemma 18.** *With probability $1 - \delta/k$, $\mathrm{cost}(P_i^*, \hat{c}_i) \leq (1 + c\alpha)\mathrm{OPT}_i$ for $c = \frac{7+10\alpha-10\alpha^2}{(1-\alpha)(1-2\alpha)}$.*

We now formalize our main cost bound, success probability and run-time guarantees in theorem 19.

**Theorem 19.** *There is an algorithm for $k$-medians clustering such that given a data set $P$ and a labelling $(P_1, \ldots, P_k)$ with error rate $\alpha < 1/2$, it outputs a set of centers $\widehat{C} = (\widehat{c}_1, \ldots, \widehat{c}_k)$*

such that $\sum_{i \in k} \text{cost}(P_i^*, \widehat{c_i}) \leq (1 + c\alpha)\text{OPT}_i$ for $c = \frac{7 + 10\alpha - 10\alpha^2}{(1-\alpha)(1-2\alpha)}$, and does so in time $O\left(\frac{1}{1-2\alpha} md \log^3\left(\frac{m}{\alpha}\right) \log\left(\frac{k \log(k/\delta)}{(1-2\alpha)\delta}\right) \log\left(\frac{k}{\delta}\right)\right)$.

*Proof.* We see from lemma 18 that by applying our subroutine for 1-median clustering on each labelled partition $P_i$, we get a center $\widehat{c_i}$ with the promise that with probability $1 - \frac{\delta}{k}$, $\text{cost}(P_i^*, \widehat{c_i}) = (1 + c\alpha)\text{OPT}_i$. By the union bound, it follows that with probability $1 - \delta$, $\sum_{i \in [k]} \text{cost}(P_i^*, \widehat{c_i}) \leq \sum_{i \in k}(1 + c\alpha)\text{OPT}_i = (1 + c\alpha)\text{OPT}$. Since $P = P_1^* \cup \cdots \cup P_k^*$, it follows that $\text{cost}(P, \hat{C}) = (1 + c\alpha)\text{OPT}$.

The time taken to execute the 1-median clustering subroutine on partition $P_i$ is $R(m_i d + O(m_i \log m_i) + O(m_i d \log^3(m_i/\gamma) \log(Rk/\delta)) + m_i d)$. This is because we have $R$ iterations, in each of which we first compute the distances of all $m_i$ points from the sampled point $x$ in time $m_i d$, followed by sorting the $m_i$ many points by their distances in time $O(m_i \log m_i)$, followed by $O(\log(Rk/\delta))$ many iterations of the median computation for the clipped sets (wherein we appeal to corollary 10), followed by a calculation of the 1-median clustering cost achieved in time $m_i d$. We recall that we set $R = O\left(\frac{1}{1-2\alpha} \log \frac{k}{\delta}\right)$. Further, we note that the expression for the upper bound on the time complexity is convex in $m_i$, so if we were to denote the value of this expression on a set of size $m_i$ by $T(m_i)$, it follows that $\sum_{i \in [k]} T(m_i) \leq T\left(\sum_{i \in [k]} m_i\right) = T(m)$. Putting everything together, we get that the net time complexity is $O\left(\frac{1}{(1-2\alpha)} md \log^3\left(\frac{m}{\alpha}\right) \log\left(\frac{k \log(k/\delta)}{(1-2\alpha)\delta}\right) \log\left(\frac{k}{\delta}\right)\right)$. $\square$

## 4 EXPERIMENTS

In this section, we evaluate algorithm 1 and algorithm 2 on real-world datasets. Our experiments were done on a i9-12900KF processor with 32GB RAM. For all experiments, we fix the number of points to be allocated $k = 10$, and report the average and the standard deviation error of the clustering cost over 20 independent runs [1].

**Datasets.** We test the algorithms on the testing set of the **CIFAR-10** dataset (Krizhevsky et al., 2009) ($m = 10^4, d = 3072$), the **PHY** dataset from KDD Cup 2004 (KDD Cup 2004), and the **MNIST** dataset (Deng, 2012) ($m = 1797, d = 64$). For the PHY dataset, we take $m = 10^4$ random samples to form our dataset ($d = 50$).

**Predictor description.** For each dataset, we create a predictor by first finding good $k$-means and $k$-medians solutions. Specifically, for $k$-means we initialize by `kmeans++` and then run Lloyd's algorithm until convergence. For $k$-medians, we use the "alternating" heuristic (Park & Jun, 2009) of the $k$-medoids problem to find the center of each cluster. In both settings, we use the label given to each point by the $k$-means and $k$-medians solutions to form the optimal partition $(P_1^*, \ldots, P_{10}^*)$ (recall we set $k = 10$). In order to test the algorithms' performance under different error rates of the predictor, for each cluster $i$, we change the labels of the $\alpha m_i$ points closest to the mean (or median) to that of a random center. For every dataset, we generate the set of corrupted labels $(P_1, \ldots, P_{10})$ for $\alpha$ from 0.1 to 0.5. Furthermore, we use the same set of optimal partition $(P_1^*, \ldots, P_{10}^*)$ across all instances of the algorithms. By fixing the optimal partition, we can investigate the effects of increasing $\alpha$ on the clustering cost.

**Guessing the error rate.** Note that in most situations, we will not have access to the error rate $\alpha$ and must try out different guesses of $\alpha$ then return the clustering with the best cost. For algorithm 1, algorithm 2, and the $k$-medians algorithm of Ergun et al. (2022), we iterate over 15 possible value of $\alpha$ uniformly spanning the interval $[0.1, 0.5]$. For the $k$-means algorithm of Ergun et al. (2022), the algorithm is defined for $\alpha < 1/5$ (not to be confused with the assumption that $\alpha < 1/7$ for the bound on the clustering cost). Thus, the range is $[0.1, 1/5]$ for the algorithm.

**Baselines.** We report the clustering costs of the initial **optimal** $k$-means and $k$-medians solution $(P_1^*, \ldots, P_{10}^*)$ along with that of the naive approach of taking the average and geometric median of each group returned by the **predictor**, e.g., returning $(\overline{P_1}, \ldots, \overline{P_{10}})$ for $k$-means. The two baselines help us see how much the clustering cost increases for different error rate $\alpha$. The clustering cost

---

[1]The repository is hosted at github.com/thydnguyen/LA-Clustering.

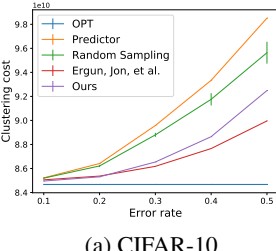 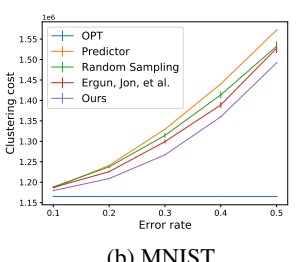 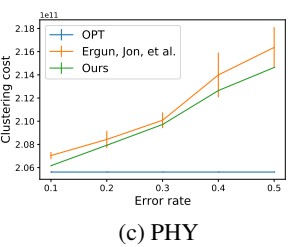

(a) CIFAR-10        (b) MNIST        (c) PHY

Figure 1: Experimental comparison of algorithm 1 with prior work and baselines for $k$-Means

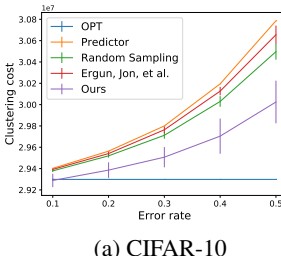 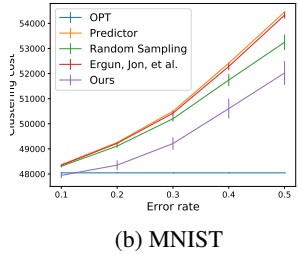 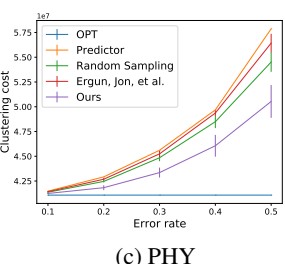

(a) CIFAR-10        (b) MNIST        (c) PHY

Figure 2: Experimental comparison of algorithm 2 with prior work and baselines for $k$-Medians

of the algorithm without corruption can also be seen as a lower bound on the cost of the learning-augmented algorithms. Following Ergun et al. (2022), we use **random sampling** as another baseline. We first randomly select a $q$-fraction of points from each cluster for $q$ varied from $1\%$ to $50\%$. Then, we compute the means and the geometric medians of the sampled points to calculate the clustering cost. Finally, we return the clustering corresponding to the value of $q$ with the best cost.

We use the implementation provided in Ergun et al. (2022) for their $k$-means algorithm. Although both our $k$-medians algorithm and the algorithm in Ergun et al. (2022) use the approach in Cohen et al. (2016) as the subroutine to compute the geometric median in nearly linear time, we use Weiszfeld's algorithm as implemented in Pillutla et al. (2022), a well-known method to compute the geometric medians, for the $k$-medians algorithms. To generate the predictions, we use Pedregosa et al. (2011); Scikit-Learn-Contrib (2021) for the implementations of the $k$-means and $k$-medoids algorithms, and the code provided in Ergun et al. (2022) for the implementation of their $k$-means algorithm.

For algorithm 2, we can treat the number of rounds $R$ as a hyperparameter. We set $R = 1$; as shown below, this is already enough to achieve a good performance compared to the other approaches.

## 4.1 RESULTS

In Figure 1, we omit the Sampling and the Prediction approach for the PHY dataset as they have much larger clustering cost than ours and the $k$-means algorithm in Ergun et al. (2022). For the CIFAR-10 dataset, we observe that the approach in Ergun et al. (2022) has slightly better clustering costs as $\alpha$ increases. For the MNIST dataset, our approach has slightly improved costs across all values of $\alpha$. For the PHY dataset, observe that algorithm 1 is comparable to the Ergun et al. (2022).

In summary, the mean clustering cost of the two learning-augmented algorithms are similar across the datasets. It is important to note that our algorithm achieves similar clustering cost to that of Ergun et al. (2022) without any variance as it is a deterministic technique.

Figure 2 shows that our our k-medians algorithm has the best clustering cost across all the datasets. We also observe that the sampling approach outperforms the approach of Ergun et al. (2022) for the CIFAR-10 and the MNIST datasets. This is expected since the latter algorithm sample a random subset of a fixed size in each cluster while the baseline approach samples subsets of different sizes and uses the one with the best cost.

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

## A  APPENDIX

### A.1  MISSING PROOFS FOR $k$-MEANS

**Lemma 3.** *For any partition $J_1 \cup J_2$ of a set $J \subset \mathbb{R}$ of size $n$, if $|J_1| \geq (1-\lambda)n$, then $|\overline{J} - \overline{J}_1|^2 \leq \frac{\lambda}{(1-\lambda)n} \operatorname{cost}(J, \overline{J})$.*

*Proof.* We know $|J_1| = (1-x)n, |J_2| = xn$ for some $x \leq \lambda$. It follows that
$$\overline{J} = (1-x)\overline{J}_1 + x\overline{J}_2$$
$$\Rightarrow |\overline{J} - \overline{J}_1| = x|\overline{J}_2 - \overline{J}_1|$$
$$\text{and } |\overline{J} - \overline{J}_2| = (1-x)|\overline{J}_2 - \overline{J}_1|$$
$$\Rightarrow |\overline{J} - \overline{J}_2| = \frac{1-x}{x}|\overline{J} - \overline{J}_1|. \tag{2}$$
We now observe that we can write
$$\operatorname{cost}(J, \overline{J}) = \operatorname{cost}(J_1, \overline{J}) + \operatorname{cost}(J_2, \overline{J}).$$
and recall the identity
$$\operatorname{cost}(J_b, \overline{J}) = \operatorname{cost}(J_b, \overline{J}_b) + |J_b| \cdot |\overline{J} - \overline{J}_b|^2$$
for $b \in \{0, 1\}$. It then follows that
$$\operatorname{cost}(J, \overline{J}) \geq |J_1| \cdot |\overline{J} - \overline{J}_1|^2 + |J_2| \cdot |\overline{J} - \overline{J}_2|^2$$
$$= (1-x)n|\overline{J} - \overline{J}_1|^2 + xn|\overline{J} - \overline{J}_2|^2$$
$$= (1-x)n|\overline{J} - \overline{J}_1|^2 + \frac{(1-x)^2 n}{x}|\overline{J} - \overline{J}_1|^2$$
$$= \frac{(1-x)n}{x}|\overline{J} - \overline{J}_1|^2$$
$$\geq \frac{(1-\lambda)n}{\lambda}|\overline{J} - \overline{J}_1|^2$$
$$\Rightarrow |J - J_1|^2 \leq \frac{\lambda}{(1-\lambda)n} \operatorname{cost}(J, \overline{J}).$$
$\square$

**Lemma 4.** *For all $i \in [k], j \in [d]$, let $\omega'_{i,j}$ be the collection of all subsets of $(1-\alpha)m_i$ points in $P_{i,j}$. Then*
$$\operatorname{cost}(I_{i,j}, \overline{I_{i,j}}) = \min_{Z' \in \omega'_{i,j}} \operatorname{cost}(Z', \overline{Z'}).$$

*Proof.* Suppose $I'_{i,j} = \operatorname{argmin}_{Z' \in \omega'_{i,j}} \operatorname{cost}(Z', \overline{Z'})$. If $I'_{i,j} \in \omega_{i,j}$ then we are done since we know:
$$\operatorname{cost}(I_{i,j}, \overline{I_{i,j}}) = \min_{Z \in \omega_{i,j}} \operatorname{cost}(Z, \overline{Z})$$
If $I'_{i,j} \notin \omega_{i,j}$, let $a$ and $b$ be the minimum point and maximum points in $I'_{i,j}$. We know there exists a point $p \in P_{i,j} \cap (a, b)$ such that $p \notin I'_{i,j}$. If $|I'_{i,j}| = 2$, then we have a contradiction since
$$\operatorname{cost}(I'_{i,j}, \overline{I'_{i,j}}) = (b-a)^2/2 > (b-p)^2/2 = \operatorname{cost}(\{b, p\}, \overline{\{b, p\}})$$
If $|I'_{i,j}| \geq 3$, we know either $a$ or $b$ is the furthest point from $\overline{I'_{i,j} \setminus \{a, b\}}$ in the interval $[a, b]$. Suppose $a$ is such a point, consider $K_{i,j} = (I'_{i,j} \setminus a) \cup p$. We have the following identity,
$$\operatorname{cost}(K_{i,j} \setminus p, p) = \operatorname{cost}(K_{i,j} \setminus \{p, b\}, p) + |p - b|^2$$
$$= \operatorname{cost}(I'_{i,j} \setminus \{a, b\}, p) + |p - b|^2$$
$$= \operatorname{cost}(I'_{i,j} \setminus \{a, b\}, \overline{I'_{i,j} \setminus a, b}) + |I'_{i,j} \setminus \{a, b\}| \cdot |p - \overline{I'_{i,j} \setminus \{a, b\}}|^2 + |p - b|^2$$
$$< \operatorname{cost}(I'_{i,j} \setminus \{a, b\}, \overline{I'_{i,j} \setminus a, b}) + |I'_{i,j} \setminus \{a, b\}| \cdot |a - \overline{I'_{i,j} \setminus \{a, b\}}|^2 + |a - b|^2$$
$$= \operatorname{cost}(I'_{i,j} \setminus \{a, b\}, a) + |a - b|^2$$
$$= \operatorname{cost}(I'_{i,j} \setminus \{a\}, a).$$

For the inequality, we used the fact that $a$ is the furthest point from $\overline{I'_{i,j} \setminus \{a,b\}}$ in the interval $[a,b]$, and $q \in (a,b)$. We have,

$$
\begin{aligned}
\text{cost}(K_{i,j}, \overline{K_{i,j}}) &= \frac{1}{(1-\alpha)m_i} \sum_{y_1,y_2 \in K_{i,j}} |y_1 - y_2|^2 \\
&= \frac{1}{(1-\alpha)m_i} \left( \sum_{y_1,y_2 \in K_{i,j} \setminus p} |y_1 - y_2|^2 + \text{cost}(K_{i,j} \setminus p, p) \right) \\
&= \frac{1}{(1-\alpha)m_i} \left( \sum_{y_1,y_2 \in I'_{i,j} \setminus a} |y_1 - y_2|^2 + \text{cost}(I'_{i,j} \setminus a, p) \right) \\
&< \frac{1}{(1-\alpha)m_i} \left( \sum_{y_1,y_2 \in I'_{i,j} \setminus a} |y_1 - y_2|^2 + \text{cost}(I'_{i,j} \setminus a, a) \right) \\
&= \frac{1}{(1-\alpha)m_i} \sum_{y_1,y_2 \in I'_{i,j}} |y_1 - y_2|^2 \\
&= \text{cost}(I'_{i,j}, \overline{I'_{i,j}})
\end{aligned}
$$

Hence, $\text{cost}(K_{i,j}, \overline{K_{i,j}}) < \text{cost}(I'_{i,j}, \overline{I'_{i,j}})$ and we have a contradiction. $\qquad\square$

**Lemma 5.** *The following bound holds:*

$$
|\overline{I_{i,j}} - \overline{Q_{i,j}}|^2 \leq \frac{4\alpha}{1 - 2\alpha} \frac{\text{cost}(Q_{i,j}, \overline{Q_{i,j}})}{|Q_i|}.
$$

*Proof.* Consider the set $S_{i,j} = \{(\overline{Q_{i,j}} - q)^2 : q \in Q_{i,j}\}$. Let $V_{i,j}$ be a subset of size $(1-\alpha)m$ drawn uniformly at random from $Q_{i,j}$. Since the sample mean is an unbiased estimator for the population mean, we know

$$
\frac{1}{(1-\alpha)m_i} \mathbb{E}\left[ \sum_{q \in V_{i,j}} (\overline{Q_{i,j}} - q)^2 \right] = \overline{S_{i,j}} = \frac{\text{cost}(Q_{i,j}, \overline{Q_{i,j}})}{|Q_i|}.
$$

We also know that,

$$
\mathbb{E}\left[ \sum_{q \in V_{i,j}} (\overline{Q_{i,j}} - q)^2 \right] = \mathbb{E}\left[ \text{cost}(V_{i,j}, \overline{Q_{i,j}}) \right] \geq \mathbb{E}\left[ \text{cost}(V_{i,j}, \overline{V_{i,j}}) \right] \geq \text{cost}(I_{i,j}, \overline{I_{i,j}}),
$$

where we used the fact that $I_{i,j}$ is a subset of size $(1 - \alpha)|P_i|$ with minimum 1-means clustering cost (lemma 4). Thus, we have

$$
\text{cost}(I_{i,j}, \overline{I_{i,j}}) \leq \frac{(1-\alpha)m_i}{|Q_i|} \text{cost}(Q_{i,j}, \overline{Q_{i,j}}).
$$

Now, in the notation of lemma 3, we set $J = I_{i,j}$ and $J_1 = I_{i,j} \cap P^*_{i,j}$. Since we have that $|I_{i,j}| = (1-\alpha)m_i$ and $|I_{i,j} \cap P^*_{i,j}| = |I_{i,j} \cap Q_{i,j}| = (1 - \frac{|P_{i,j} \setminus Q_{i,j}|}{1 - m_i})(1-\alpha)m_i$, we can set $\lambda = \frac{|P_{i,j} \setminus Q_{i,j}|}{1 - m_i}$, and get that

$$
\begin{aligned}
|\overline{I_{i,j}} - \overline{I_{i,j} \cap Q_{i,j}}|^2 &\leq \frac{|P_{i,j} \setminus Q_{i,j}| \, \text{cost}(I_{i,j}, \overline{I_{i,j}})}{((1-\alpha)m_i - |P_{i,j} \setminus Q_{i,j}|)(1-\alpha)m_i} \\
&\leq \frac{|P_{i,j} \setminus Q_{i,j}| \, \text{cost}(Q_{i,j}, \overline{Q_{i,j}})}{((1-\alpha)m_i - |P_{i,j} \setminus Q_{i,j}|)|Q_i|} \\
&\leq \frac{\alpha \, \text{cost}(Q_{i,j}, \overline{Q_{i,j}})}{(1 - 2\alpha)|Q_i|},
\end{aligned}
$$

where we use the fact that $|P_{i,j} \setminus Q_{i,j}| \le \alpha m_i$. Also, by lemma 3,

$$|\overline{Q_{i,j}} - \overline{I_{i,j} \cap Q_{i,j}}|^2 \le \frac{\alpha m_i \, \mathrm{cost}(Q_{i,j}, \overline{Q_{i,j}})}{(|Q_{i,j}| - \alpha m_i)|Q_{i,j}|} \le \frac{\alpha \, \mathrm{cost}(Q_{i,j}, \overline{Q_i})}{(1 - 2\alpha)|Q_i|}$$

We conclude the proof by noting that

$$|\overline{I_{i,j}} - \overline{Q_{i,j}}|^2 \le 2|\overline{I_{i,j}} - \overline{I_{i,j} \cap Q_{i,j}}|^2 + 2|\overline{I_{i,j} \cap Q_{i,j}} - \overline{Q_{i,j}}|^2.$$

$\square$

**Lemma 6.** *The following bound holds:*

$$|\overline{P^*_{i,j}} - \overline{I_{i,j}}|^2 \le \mathrm{cost}(P^*_{i,j}, \overline{P^*_{i,j}}) \left( \frac{\alpha}{1 - \alpha} + \frac{4\alpha}{(1 - 2\alpha)(1 - \alpha)} \right) / m^*_i$$

*Proof.* By eq. (2),

$$|\overline{P^*_{i,j}} - \overline{P^*_{i,j} \setminus Q_{i,j}}|^2 = \frac{(1 - z)^2}{z^2} |\overline{P^*_{i,j}} - \overline{Q_{i,j}}|^2,$$

where $z = \frac{|P^*_{i,j} \setminus Q_{i,j}|}{|P^*_{i,j}|} \le \alpha$. We have

$$
\begin{aligned}
& \mathrm{cost}(P^*_{i,j}, \overline{P^*_{i,j}}) \\
& = \mathrm{cost}(P^*_{i,j} \setminus Q_{i,j}, \overline{P^*_{i,j}}) + \mathrm{cost}(Q_{i,j}, \overline{P^*_{i,j}}) \\
& = \mathrm{cost}(P^*_{i,j} \setminus Q_{i,j}, \overline{P^*_{i,j} \setminus Q_{i,j}}) + z m^*_i |\overline{P^*_{i,j}} - \overline{P^*_{i,j} \setminus Q_{i,j}}|^2 + \mathrm{cost}(Q_{i,j}, \overline{Q_{i,j}}) \\
& \quad + (1 - z) m^*_i |\overline{P^*_{i,j}} - \overline{Q_{i,j}}|^2 \\
& = \frac{1 - z}{z} m^*_i |\overline{P^*_{i,j}} - \overline{Q_{i,j}}|^2 + \mathrm{cost}(P^*_{i,j} \setminus Q_{i,j}, \overline{P^*_{i,j} \setminus Q_{i,j}}) + \mathrm{cost}(Q_{i,j}, \overline{Q_{i,j}}) \\
& \ge \frac{1 - \alpha}{\alpha} m^*_i |\overline{P^*_{i,j}} - \overline{Q_{i,j}}|^2 + \mathrm{cost}(Q_{i,j}, \overline{Q_{i,j}}).
\end{aligned}
$$

Applying lemma 5, we have

$$\mathrm{cost}(P^*_{i,j}, \overline{P^*_{i,j}}) \ge \frac{1 - \alpha}{\alpha} m^*_i |\overline{P^*_{i,j}} - \overline{Q_{i,j}}|^2 + \frac{1 - 2\alpha}{4\alpha} \cdot (1 - \alpha) m^*_i |\overline{I_{i,j}} - \overline{Q_{i,j}}|^2.$$

By Cauchy-Schwarz,

$$
\begin{aligned}
& \left( |\overline{P^*_{i,j}} - \overline{Q_{i,j}}| + |\overline{I_{i,j}} - \overline{Q_{i,j}}| \right)^2 \\
& \le \left( \frac{\alpha}{1 - \alpha} + \frac{4\alpha}{(1 - 2\alpha)(1 - \alpha)} \right) \\
& \quad \left( \frac{1 - \alpha}{\alpha} m^*_i |\overline{P^*_{i,j}} - \overline{Q_{i,j}}|^2 + \frac{1 - 2\alpha}{4\alpha} \cdot (1 - \alpha) m^*_i |\overline{I_{i,j}} - \overline{Q_{i,j}}|^2 \right) / m^*_i \\
& \le \mathrm{cost}(P^*_{i,j}, \overline{P^*_{i,j}}) \left( \frac{\alpha}{1 - \alpha} + \frac{4\alpha}{(1 - 2\alpha)(1 - \alpha)} \right) / m^*_i
\end{aligned}
$$

We conclude the proof by the fact that $|\overline{P^*_{i,j}} - \overline{I_{i,j}}|^2 \le \left( |\overline{P^*_{i,j}} - \overline{Q_{i,j}}| + |\overline{I_{i,j}} - \overline{Q_{i,j}}| \right)^2$. $\square$

**Theorem 7.** *Algorithm 1 is a deterministic algorithm for $k$-means clustering such that given a data set $P \in \mathbb{R}^{m \times d}$ and a partition $(P_1, \ldots, P_k)$ with error rate $\alpha < 1/2$, it outputs a $\left( 1 + \left( \frac{\alpha}{1 - \alpha} + \frac{4\alpha}{(1 - 2\alpha)(1 - \alpha)} \right) \right)$-approximation in time $O\left( dm \log m \right)$.*

*Proof.* Recall that the $k$-means clustering cost can be written as the sums of the clustering cost in each dimension. For every $i \in [k]$, we have

$$
\begin{aligned}
\sum_{i \in [k]} \mathrm{cost}(P_i^*, \{\widehat{c}_j\}_{j=1}^k) &\leq \sum_{i \in [k]} \mathrm{cost}(P_i^*, \widehat{c}_i) \\
&= \sum_{i \in [k]} \sum_{j \in [d]} \mathrm{cost}(P_{i,j}^*, \widehat{c}_{i,j}) \\
&= \sum_{i \in [k]} \sum_{j \in [d]} \mathrm{cost}(P_{i,j}^*, \overline{P_{i,j}^*}) + m_i^* |\widehat{c}_{i,j} - \overline{P_{i,j}^*}| \\
&= \sum_{i \in [k]} \sum_{j \in [d]} \mathrm{cost}(P_{i,j}^*, \overline{P_{i,j}^*}) + m_i^* |\overline{I_{i,j}} - \overline{P_{i,j}^*}| \\
&\leq \sum_{i \in [k]} \sum_{j \in [d]} \left( 1 + \left( \frac{\alpha}{1 - \alpha} + \frac{4\alpha}{(1 - 2\alpha)(1 - \alpha)} \right) \right) \mathrm{cost}(P_{i,j}^*, \overline{P_{i,j}^*}) \\
&= \left( 1 + \left( \frac{\alpha}{1 - \alpha} + \frac{4\alpha}{(1 - 2\alpha)(1 - \alpha)} \right) \right) \sum_{i \in [k]} \mathrm{cost}(P_i^*, c_i^*).
\end{aligned}
$$

The inequality is due to lemma 6.

We analyze the runtime of algorithm 1. Notice for every $i \in [k], j \in [d]$, computing $\overline{I_{i,j}}$ involves sorting the points $P_{i,j}$, iterating from the smallest to the largest point, and taking the average of the interval in $\omega_{i,j}$ with the smallest cost. This takes $O(m_i \log m_i)$ time. Note that $\sum_{i \in [K]} m_i = m$. Thus, the total time over all $i \in [k]$ and $j \in [d]$ is $O(dm \log m)$. $\qquad\square$

**Corollary 8.** *For $\alpha \leq 1/7$, algorithm 1 achieves a clustering cost of $(1 + 7.7\alpha)$OPT.*

*Proof.* We recall that the generic guarantee for $\alpha < 1/2$ is

$$
\mathrm{cost}(P, \{\widehat{c}_1, \ldots, \widehat{c}_k\}) \leq \left( 1 + \left( \frac{\alpha}{1 - \alpha} + \frac{4\alpha}{(1 - 2\alpha)(1 - \alpha)} \right) \right) \mathrm{OPT}.
$$

We see that for $\alpha < 1/7$, $\frac{\alpha}{1-\alpha} \leq \frac{7\alpha}{6}$, and $\frac{4\alpha}{(1-2\alpha)(1-\alpha)} \leq \frac{49 \cdot 4\alpha}{30}$, so in sum the net approximation factor is $1 + 7.7\alpha$. $\qquad\square$

### A.2 MISSING PROOFS FOR $k$-MEDIANS

**Lemma 12.** *With probability $\frac{1-2\alpha}{2}$, $\|x - c_i^*\| \leq 2\mathrm{OPT}_i/m_i$.*

*Proof.* We observe that $\mathrm{cost}(P_i \cap P_i^*, c_i^*) \leq \mathrm{OPT}_i$. It follows that $\mathbb{E}_{x \sim P_i \cap P_i^*}[\|x - c_i^*\|] \leq \frac{\mathrm{OPT}_i}{|P_i \cap P_i^*|} \leq \frac{\mathrm{OPT}_i}{(1-\alpha)m_i}$. By Markov's inequality,

$$
\begin{aligned}
\Pr\left( \|x - c_i^*\| > (1 + \epsilon) \cdot \frac{\mathrm{OPT}_i}{(1 - \alpha)m_i} \Big| x \in P_i \cap P_i^* \right) &\leq \frac{1}{1 + \epsilon} \\
\Rightarrow \frac{\Pr\left( \|x - c_i^*\| \leq \frac{(1+\epsilon)\mathrm{OPT}_i}{(1-\alpha)m_i} \wedge x \in P_i \cap P_i^* \right)}{P(x \in P_i \cap P_i^*)} &\geq \frac{\epsilon}{1 + \epsilon} \\
\Pr\left( \|x - c_i^*\| \leq \frac{(1 + \epsilon)\mathrm{OPT}_i}{(1 - \alpha)m_i} \right) &\geq \frac{\epsilon}{1 + \epsilon} P(x \in P_i \cap P_i^*) \\
&\geq \frac{\epsilon(1 - \alpha)}{1 + \epsilon}
\end{aligned}
$$

To get the stated bound we set $\epsilon = 1 - 2\alpha$. $\qquad\square$

**Lemma 14.** *Conditioned on $\mathcal{E}$, $\mathrm{cost}(P_i \setminus P_i', \widehat{c}_i^j) \leq (1 + 5\alpha)\mathrm{OPT}_i$.*

We first define some notation for the sets of false positive and false negative points that occur in our proof for lemma 14, and prove a technical lemma relating the sets $P_i \cap P_i^*$ and $P_i \backslash P_i'$.

**Definition 20.** *We make the following definitions:*

1. *Let $\mathcal{E}_1$ denote the event that $\|x - c_i^*\| \leq 2\mathrm{OPT}_i/n$.*

2. *Let $A$ denote the set of false negatives, i.e. $P_i^* \cap P_i'$.*

3. *Let $B$ denote the set of false positives, i.e. $P_i \backslash (P_i' \cup P_i^*)$.*

To bound the clustering cost of $P_i \backslash P_i'$, in terms of the cost of $P_i \cap P_i'$, we first relate these two sets in terms of the false positives $B$ and the false negatives $A$.

**Lemma 21.** *We can write $P_i \cap P_i^* = ((P_i \backslash P_i') \backslash B) \cup A$ (see definition 20 for the definitions of $A$ and $B$).*

*Proof.* To see this we observe that

$$P_i \backslash P_i' = ((P_i \backslash P_i') \cap P_i^*) \cup ((P_i \backslash P_i') \backslash P_i^*)$$
$$= ((P_i \backslash P_i') \cap P_i^*) \cup B$$
$$\Rightarrow (P_i \backslash P_i') \cap P_i^* = (P_i \backslash P_i') \backslash B.$$

We also have that

$$P_i \cap P_i^* = ((P_i \cap P_i^*) \cap P_i') \cup ((P_i \cap P_i^*) \backslash P_i')$$
$$\Rightarrow ((P_i \cap P_i^*) \backslash P_i') = (P_i \cap P_i^*) \backslash ((P_i \cap P_i^*) \cap P_i')$$
$$= (P_i \cap P_i^*) \backslash A.$$

Since $(P_i \cap P_i^*) \backslash P_i' = (P_i \backslash P_i') \cap P_i^*$, we can identify the left hand sides in the last two displays and write

$$(P_i \cap P_i^*) \backslash A = (P_i \backslash P_i') \backslash B$$
$$\Rightarrow P_i \cap P_i^* = ((P_i \backslash P_i') \backslash B) \cup A.$$

wherein we use that $A = (P_i \cap P_i^*) \cap P_i'$. $\qquad\square$

We can now formalize our main argument showing that the clipped data set $P_i \backslash P_i'$ has a clustering cost close to that of the true cluster $P_i^*$.

*Proof of lemma 14.* By lemma 21, we first observe that

$$\mathrm{cost}(P_i \setminus P_i', c_i^*) = \mathrm{cost}((P_i \cap P_i^*), c_i^*) - \mathrm{cost}(A, c_i^*) + \mathrm{cost}(B, c_i^*),$$

where $A$ and $B$ are defined as in definition 20. Again by lemma 21, $P_i \setminus P_i' = (P_i \cap P_i^*) \setminus A \cup B$, $A \subset P_i \cap P_i^*$ and $B \cap (P_i \cap P_i^*) = \emptyset$, it follows that

$$|P_i \setminus P_i'| = |P_i \cap P_i^*| - |A| + |B|.$$

Further, we know that $|P_i \setminus P_i'| \leq (1 - \alpha)|P_i|$ and $|P_i \cap P_i^*| \geq (1 - \alpha)|P_i|$. It follows that $|B| \leq |A| \leq \alpha n$. Therefore, for every false positive $p \in B$, we can assign a unique corresponding false negative $n_p \in A$ arbitrarily. We observe that every point in $A$ is farther from $x$ than every point in $B$, and so we can write

$$\|n_p - c_i^*\| \geq \|n_p - x\| - \|x - c_i^*\|$$
$$\geq \|p - x\| - \|x - c_i^*\|$$
$$\geq \|p - c_i^*\| - 2\|x - c_i^*\|$$
$$\geq \|p - c_i^*\| - \frac{4\mathrm{OPT}_i}{n}$$
$$\Rightarrow \|p - c_i^*\| \leq \|n_p - c_i^*\| + \frac{4\mathrm{OPT}_i}{n}.$$

It follows that

$$\text{cost}(B, c_i^*) = \sum_{p \in B} \|p - c_i^*\|$$

$$\leq \sum_{p \in B} \|n_p - c_i^*\| + \frac{4\text{OPT}_i}{n}$$

$$\leq \text{cost}(A, c_i^*) + 4\alpha \text{OPT}_i.$$

Returning to our expression for $\text{cost}(P_i \setminus P_i', m_i^*)$, we get that

$$\text{cost}(P_i \setminus P_i', c_i^*) = \text{cost}((P_i \cap P_i^*) \setminus A, c_i^*) + \text{cost}(B, c_i^*)$$

$$= \text{cost}((P_i \cap P_i^*), c_i^*) - \text{cost}(A, c_i^*) + \text{cost}(B, c_i^*)$$

$$\leq \text{cost}((P_i \cap P_i^*), c_i^*) + 4\alpha \text{OPT}_i$$

$$\leq (1 + 4\alpha)\text{OPT}_i.$$

It follows that the optimal clustering cost for the set $P_i \setminus P_i'$ is at most $(1 + 4\alpha)\text{OPT}_i$, and hence that $\text{cost}(P_i \setminus P_i', \widehat{c}_i^j) \leq (1 + \gamma)(1 + 4\alpha)\text{OPT}_i \leq (1 + 5\alpha)\text{OPT}_i$, for suitably small $\gamma \leq \frac{\alpha}{1+4\alpha}$. $\quad\square$

**Lemma 15.** *For $R = O\left(\frac{1}{(1-2\alpha)} \log\left(\frac{2k}{\delta}\right)\right)$ many repetitions, with probability at least $1 - \frac{\delta}{2k}$, we have that $\text{cost}(P_i \setminus P_i', \widehat{c}_i) \leq (1 + 5\alpha)\text{OPT}_i$.*

*Proof.* The probability $\mathcal{E}_1$ not holding for some $\widehat{c}_i^j$ is at most $(1 - 2\alpha)/2$. The probability of $\mathcal{E}_1$ not holding for any of the $\widehat{c}_i^j$ is $(1 - (1 - 2\alpha)/2)^R$. It follows that for $R = \frac{2}{1-2\alpha} \ln\left(\frac{2k}{\delta}\right)$, the probability of $\mathcal{E}_1$ not holding for any of the $m_i^j$ is at most

$$(1 - (1 - 2\alpha)/2)^R \leq \exp(-(1 - 2\alpha)/2)^R$$

$$\leq \exp\left(-\ln\left(2k/\delta\right)\right)$$

$$\leq \frac{\delta}{2k}.$$

It follows that with probability $1 - \frac{\delta}{2k}$, $\mathcal{E}_1$ holds for some $\widehat{c}_i^j$ and consequently by the union bound $\text{cost}(P_i \setminus P_i', \widehat{c}_i) \leq (1 + 5\alpha)\text{OPT}_i$ holds with probability $1 - \frac{\delta}{k}$. $\quad\square$

**Lemma 16.** *If $\text{cost}(P_i \setminus P_i', \widehat{c}_i) \leq (1 + 5\alpha)\text{OPT}$, then $\|\widehat{c}_i - c_i^*\| \leq \frac{2+5\alpha}{(1-2\alpha)} \frac{\text{OPT}_i}{n}$.*

*Proof.* By the reverse triangle inequality we have that for every point $p \in P_i^* \cap (P_i \setminus P_i')$, $\|\widehat{c}_i - p\| \geq \|\widehat{c}_i - c_i^*\| - \|p - c_i^*\|$. Summing up across p, we get

$$\sum_{p \in P_i^* \cap (P_i \setminus P_i')} \|\widehat{c}_i - p\| \geq |P_i^* \cap (P_i \setminus P_i')| \cdot \|\widehat{c}_i - c_i^*\| - \sum_{p \in P_i^* \cap (P_i \setminus P_i')} \|p - c_i^*\|$$

$$(1 + 5\alpha)\text{OPT}_i \geq |P_i^* \cap (P_i \setminus P_i')| \cdot \|\widehat{c}_i - c_i^*\| - \text{OPT}_i$$

$$\Rightarrow |P_i^* \cap (P_i \setminus P_i')| \cdot \|\widehat{c}_i - c_i^*\| \leq ((1 + 5\alpha) + 1)\text{OPT}_i$$

$$\Rightarrow \|\widehat{c}_i - c_i^*\| \leq \frac{(2 + 5\alpha)\text{OPT}_i}{(1 - 2\alpha)m_i}.$$

$\square$

**Lemma 17.** *With probability $1 - \delta/k$, $\text{cost}(P_i \cap P_i^*, \widehat{c}_i) \leq \text{cost}(P_i \cap P_i^*, c_i^*) + \frac{(5\alpha + 10\alpha^2)\text{OPT}_i}{1 - 2\alpha}$.*

*Proof.* From corollary 10, we know that with probability $1 - \frac{\delta}{2k}$, the following bound holds:

$$\text{cost}(P_i \setminus P_i', \widehat{c}_i) \leq (1 + \gamma)\text{cost}(P_i \setminus P_i', c_i'),$$

where $\gamma \leq \frac{\alpha}{(1+4\alpha)}$ and $c_i'$ is an optimal 1-median for $P_i \backslash P_i'$. Also, it follows by definition that $\text{cost}(P_i \backslash P_i', c_i') \leq \text{cost}(P_i \backslash P_i', c_i^*)$. Further, from lemma 15 and lemma 16 it follows that with probability $1 - \frac{\delta}{2k}$,

$$\|\widehat{c}_i - c_i^*\| \leq \frac{(2+5\alpha)\text{OPT}_i}{(1-2\alpha)m_i}.$$

By the union bound, both these events hold simultaneously with probability $1 - \frac{\delta}{k}$. Conditioning on this being the case, since $P_i \cap P_i^* = ((P_i \backslash P_i') \backslash B) \cup A$, we can write

$$
\begin{aligned}
\text{cost}(P_i \cap P_i^*, \widehat{c}_i) - \text{cost}(P_i \cap P_i^*, c_i^*) &= (\text{cost}(P_i \backslash P_i', \widehat{c}_i) - \text{cost}(P_i \backslash P_i', c_i^*)) \\
&\quad + (\text{cost}(B, c_i^*) - \text{cost}(B, \widehat{c}_i)) \\
&\quad + (\text{cost}(A, \widehat{c}_i) - \text{cost}(A, c_i^*)) \\
&\leq (1+\gamma)\text{cost}(P_i \backslash P_i', c_i') - \text{cost}(P_i \backslash P_i', c_i') \\
&\quad + |B| \cdot \|\widehat{c}_i - c_i^*\| + |A| \cdot \|\widehat{c}_i - c_i^*\| \\
&\leq \gamma \cdot \text{cost}(P_i \backslash P_i', c_i^*) + |B| \cdot \|\widehat{c}_i - c_i^*\| + |A| \cdot \|\widehat{c}_i - c_i^*\| \\
&\leq \alpha\text{OPT}_i + (\alpha m_i + \alpha m_i) \cdot \frac{(2+5\alpha)\text{OPT}_i}{(1-2\alpha)m_i} \\
&\leq \frac{\alpha + 2\alpha(2+5\alpha)\text{OPT}_i}{(1-2\alpha)} \\
&= \frac{(5\alpha + 10\alpha^2)\,\text{OPT}_i}{1-2\alpha}.
\end{aligned}
$$

$\square$

**Lemma 18.** *With probability $1 - \delta/k$, $\text{cost}(P_i^*, \hat{c}_i) \leq (1+c\alpha)\text{OPT}_i$ for $c = \frac{7+10\alpha-10\alpha^2}{(1-\alpha)(1-2\alpha)}$.*

*Proof.* We have that

$$\text{cost}(P_i^*, \widehat{c}_i) = \text{cost}(P_i^* \cap P_i, \widehat{c}_i) + \text{cost}(P_i^* \backslash P_i, \widehat{c}_i).$$

We bound the second summand as follows

$$
\begin{aligned}
\text{cost}(P_i^* \backslash P_i, \widehat{c}_i) &= \text{cost}(P_i^* \backslash P_i, c_i^*) + |P_i^* \backslash P_i| \cdot \frac{(2+5\alpha)\text{OPT}_i}{(1-2\alpha)c_i} \\
&\leq \text{cost}(P_i^* \backslash P_i, c_i^*) + \frac{\alpha(2+5\alpha)\text{OPT}_i}{(1-\alpha)(1-2\alpha)}.
\end{aligned}
$$

Bounding the first summand $\text{cost}(P_i^* \cap P_i, \widehat{c}_i)$ using the bound from above, we get

$$
\begin{aligned}
\text{cost}(P_i^*, \widehat{c}_i) &= \text{cost}(P_i^* \cap P_i, c_i^*) + \frac{(5\alpha + 10\alpha^2)\,\text{OPT}_i}{1-2\alpha} \\
&\quad + \text{cost}(P_i^* \backslash P_i, c_i^*) + \frac{\alpha(2+5\alpha)\text{OPT}_i}{(1-\alpha)(1-2\alpha)} \\
&= \text{cost}(P_i^*, c_i^*) + \frac{(5\alpha + 10\alpha^2 - 5\alpha^2 - 10\alpha^3 + 2\alpha + 5\alpha^2)\,\text{OPT}_i}{(1-\alpha)(1-2\alpha)} \\
&= \text{cost}(P_i^*, c_i^*) + \frac{(7\alpha + 10\alpha^2 - 10\alpha^3)\,\text{OPT}_i}{(1-\alpha)(1-2\alpha)}.
\end{aligned}
$$

$\square$

# B  EXPERIMENTS ON RUNTIME

In this section, we report the runtimes of our $k$-means and $k$-medians approaches and the methods in Ergun et al. (2022). We sample subsets of points from the **CIFAR-10** and the **PHY** datasets, and report the runtime (means and standard deviations) of the algorithms over 20 random runs. The

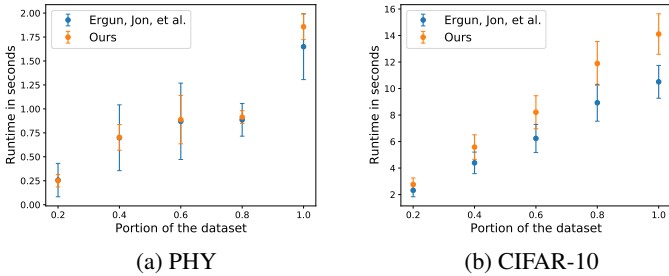

(a) PHY          (b) CIFAR-10

Figure 3: Runtime comparison of algorithm 1 with Ergun et al. (2022)

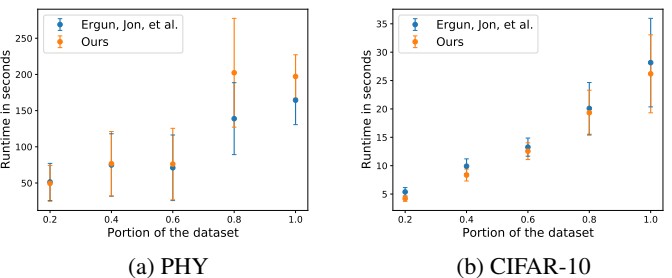

(a) PHY          (b) CIFAR-10

Figure 4: Runtime comparison of algorithm 2 with Ergun et al. (2022)

subset sizes are varied from $20\%$ to $100\%$ of the size of the datasets, $k$ is fixed at 10 and $\alpha$ is fixed at .2

For k-means, we observe in fig. 3 that the runtime of the two approaches are comparable, except for subset sizes $80\%$ and $100\%$ of CIFAR-10 where ours is slightly slower. This is expected since finding a subset of size $(1 - \alpha)m_i$ with the best clustering cost in our algorithm and computing the shortest interval containing $m_i(1 - 5\alpha)/2$ points in the approach of Ergun et al. (2022) both involve sorting the points and takes $O(m_i \log m_i)$ time.

We observe similar trends in the k-medians setting in fig. 4. This is also expected given that the runtimes of both algorithms are dominated by calls to compute the 1-median center of the filtered points in each predicted cluster.

