# OpenReview forum: "Improved Learning-augmented Algorithms for k-means and k-medians Clustering"
_ICLR.cc/2023/Conference — ICLR 2023 poster_

### Official Review · Reviewer_6m4b · 2022-10-20

**Confidence:** 4
**Clarity, Quality, Novelty And Reproducibility:** Please refer to the last question.
**Correctness:** 3
**Technical Novelty And Significance:** 3
**Empirical Novelty And Significance:** 3
**Recommendation:** 6

**Strength And Weaknesses:**

Strength

1. The technical contribution is solid. Both of the algorithms achieve a better guarantee than the previous algorithms.
2. The experimental results show that the performance of the proposed algorithm is better than the previous ones.

Weakness

1. The paper claims to preserve the time complexity of the previous approaches. However, when referring to the time complexity, the setting of this paper seems to be different than that in Ergun et al. If my understanding is correct, In Ergun et al, the time is including the time to assign each point to the closet center, which is $O(mdk)$ naively, so the authors show that even include this step, the time complexity is still $O(md \log m)$. However, for this paper, the time seems to not include this step but only output the $k$ centers. One question is that can the technique used in Algorithm 3 of Ergun et al. be applied in this paper?
2. The paper considers the time complexity of the different algorithms. However, in the experiment section, the authors seems not to list the result of the runtimes of different algorithms.

**Summary Of The Paper:**

The paper studies the $k$-means and $k$-median problem in the learning-augmented setting, where we assumed to access a predictor that provides information about the label of each point with a $(1 - \alpha)$-precision, and the approximation factor of the algorithm is measured in terms of $\alpha$. For both problems, the authors propose an algorithm that gets a better approximation. The experiments also demonstrate the advantage of these algorithms.

**Summary Of The Review:**

Overall I think it is an interesting paper. However,  the authors should clarify the issues in the time complexity(see strength and weakness).  Also, the comparison of the runtime in the experiment will make this paper more complete. Therefore, my current rating is 6.

---

> ### Author Response · Authors · 2022-11-11
> **Reply to Reviewer 6m4b**
>
> We thank the reviewer for the constructive feedback. Below, we address the reviewer’s concerns.
>
> **Time complexity**.  We affirm that our k-means algorithm can be combined with Algorithm 3 of  Ergun et al. (2021) to assign each point to a “sufficiently close” center in $O(d m \log m)$ time.
>
> Specifically, in Ergun et al. (2021), Algorithm 3 computes k approximately optimal centers for the k-means objective for a random subset of the data set using Algorithm 1. Our algorithm also similarly computes k approximately optimal centers. If one would like to obtain labels for all the input points, the remaining task amounts to the (approximate) nearest neighbor search problem where we need to find the approximate nearest center for each input point, which is a well-studied task beyond clustering. Indeed, Algorithm 3 uses a standard approach involving reducing dimensions appropriately and applying an approximate nearest neighbor data structure in low dimensions to assign each point to a “sufficiently close” center.
>
> As such, our Algorithm 1, with an improved bound on the clustering cost and the same runtime complexity as Algorithm 1 of Ergun et al. (2021), can work with Algorithm 3 of Ergun et al. (2021)  to map each point to a “sufficiently close” center in $O(d m \log m)$ time.
>
> We focus on the task of computing the approximately optimal centers as this is the central problem in clustering. Nearest neighbor search is a well-studied problem on its own and we do not have any new contribution there. It should be noted that the particular nearest neighbor data structure used in Algorithm 3 is mainly of theoretical interest and not very often used in practice. Ergun et al. (2021) did not use it in their implementation of Algorithm 3 in the experimental sections and we also follow their steps. Nonetheless, we agree with the reviewer that the above discussion is valuable to have.
>
> **Experiments on runtime**. We included experiments on the runtimes of the k-means and k-medians methods in Section B of the appendix. We observe that our methods have runtime that are largely comparable to the approaches in Ergun et al. (2021).
>
> For k-means, this is expected since finding a subset of size $(1-\alpha)m_i$ with the best clustering cost in our algorithm and computing  the shortest interval containing $m_i (1-5\alpha) / 2 $ points  in the approach of Ergun et al. (2021) both involve sorting the points and takes $O(m_i \log m_i)$ time. Since $\sum_i^k m_i =m$, the runtimes for both algorithms are $O(d m \log m)$ and the differences in the experimental runtimes are due to constant factors.
>
> The runtimes are also similar in the k-medians setting. This is also expected given that the runtimes of both algorithms are dominated by calls to compute the 1-median of the filtered points in each predicted cluster.

---

### Official Review · Reviewer_HcAi · 2022-10-20

**Confidence:** 1
**Correctness:** 3
**Technical Novelty And Significance:** 3
**Empirical Novelty And Significance:** 3
**Recommendation:** 6

**Clarity, Quality, Novelty And Reproducibility:**

The presentation quality can be improved to make it easier to be understood by readers.

Besides, it seems that the main reference (Ergun et al. 2021) has been published in ICLR2022:
https://openreview.net/forum?id=X8cLTHexYyY

It is better to update the reference list to their newest status.

**Details Of Ethics Concerns:**

N/A.

**Strength And Weaknesses:**

Strength:
This submission follows the setting in (Ergun et al. 2021), where we are given a data set in $d$-dimensional Euclidean space, and a label for each data point given by a predictor indicating what subsets of points should be clustered together.  The proposed two algorithms (Algorithm 1 and Algorithm 2) have improved bounds on the clustering cost, while preserving the time complexity of the previous approaches and removing the requirement on a lower bound on the size of each predicted cluster.

Weaknesses:
The presentation quality can be improved to make it easier to be understood by readers. For example, there is not a conclusion section in this submission. Besides, there are some typos and grammatical problems.

**Summary Of The Paper:**

This paper can be regarded as a follow-up of (Ergun et al. 2021) which is published in ICLR2022. The main contribution of this submission is to answer the question: "Is it possible to design a $k$-means and a $k$-medians algorithm that achieve (1 + $\alpha$)-approximate clustering when the predictor is not very accurate?". Moreover, the proposed two algorithms (Algorithm 1 and Algorithm 2) also have improved bounds on the clustering cost, while preserving the time complexity of the previous approaches and removing the requirement on a lower bound on the size of each predicted cluster.

[Ergun et al. 2021] Jon Ergun, Zhili Feng, Sandeep Silwal, David P. Woodruff, and Samson Zhou. Learning-augmented
k-means clustering.  In: ICLR, 2022.

**Summary Of The Review:**

This submission follows the setting in (Ergun et al. 2021), where we are given a data set in $d$-dimensional Euclidean space, and a label for each data point given by a predictor indicating what subsets of points should be clustered together. I am not familiar with the topic of this submission, and I am open to hear opinions from other reviewers.

---

> ### Author Response · Authors · 2022-11-11
> **Reply to Reviewer HcAi**
>
> We thank the reviewer for the constructive feedback. We have uploaded a rebuttal draft with an updated reference for Ergun et al. (2021) and corrected spelling and grammatical errors.

---

### Official Review · Reviewer_JvjK · 2022-10-25

**Confidence:** 4
**Correctness:** 4
**Technical Novelty And Significance:** 2
**Empirical Novelty And Significance:** Not applicable
**Recommendation:** 6

**Clarity, Quality, Novelty And Reproducibility:**

The paper is quite well written and was easy to follow. The algorithm and analysis were clean. As outlined above, the algorithm and analysis are somewhat along known lines, but the exact results are of course novel.

Reproducibility: N/A (theory paper)


**Strength And Weaknesses:**

The main strengths of the paper are:
- They analyze simple and practical algorithms for the two problems and show an approximation ratio that improves upon prior works
- The range of \alpha is considerably improved compared to prior work.

However, the paper has a few weak aspects:
- The first is wrt the motivation: the model isn't necessarily about "learning augmented" algorithms. It is much more closely related to clustering with noisy labels, which has been extensively studied (including in some recent works, e.g., in COLT 22).
- Secondly, from an algorithmic point of view, the paper is somewhat on the weak side: most of the techniques are relatively standard and it's hard to get excited about. While the overall result is clean, it is not especially surprising. Some matching lower bounds, or surprising trade-offs, or perhaps improved results with the "errors" are random, might be interesting to know about.


**Summary Of The Paper:**

The paper studies approximation algorithms for k-means and k-median clustering in the presence of "advice" in the form of an oracle that approximately provides the optimal clustering. The main assumption is that the algorithm has access to a clustering that agrees with an optimal clustering with parameter "\alpha". The parameter ensures that for every cluster, the provided clustering has a good overlap with the optimal.

Under this assumption, the authors improve upon prior work in the same model, and show a (1+O(\alpha)) approximation guarantee. Specifically, they improve prior work in terms of the dependence on \alpha, as well as the range of \alpha to which the algorithm applies.

For k-means, the authors use a "coordinate wise" algorithm that keeps track of a good subset of points (the potential "agreement" with OPT) in each cluster, and show that this must yield a good approximation ratio. For k-median the high level idea is similar, but the algorithm is randomized and it uses as a subroutine an approximate median algorithm from previous work. Both the algorithms are quite clean and practical.

**Summary Of The Review:**

In summary, the paper presents simple, practical algorithms for k-means and k-median clustering when the algorithm has access to a "noisy version" of the optimal clustering. The novelty in terms of algorithm/techniques is somewhat limited though, which is why I rate it as mildly above acceptance threshold.

---

> ### Author Response · Authors · 2022-11-11
> **Reply to Reviewer JvjK**
>
> We thank the reviewer for the constructive feedback. In the rebuttal revision, we have included references to the work on clustering with noisy information in Gamlath et al. (2022), which has related but different assumptions on the predicted cluster from our setting. Below, we address the reviewer’s concerns.
>
> **Motivation.** Although the problem of clustering with “imperfect” labels is well-studied under different names, e.g., Ergun et al. (2021) and Gamlath et al. (2022), the motivation remains the same: to overcome the computational barriers and obtain a $(1+O(\alpha))$-approximate solution efficiently. As in Ergun et al. (2021), we look beyond polynomial time as the notion of efficiency but focus on practical nearly linear-time algorithms that can scale to very large datasets. We greatly extend the range of $\alpha$ to which such algorithms apply and remove the lower bound requirement on the size of each predicted cluster.
>
> In learning augmented algorithms, there are usually two goals: 1) when the prediction is good, we obtain a very accurate solution, and 2) when the prediction is bad, we obtain a reasonable solution. Both Ergun et al. and our solutions satisfy goal 1 and for goal 2, while we do not achieve a good solution for arbitrarily bad predictions, we achieve robustness all the way to 50% error $(alpha=1/2)$ with fast runtime, which contains arguably all practical scenarios. Note that without a strict efficiency requirement, one can simply call a poly-time constant approximation algorithm to satisfy goal 2.
>
> **Technical novelty.** For k-means, our approach for removing outliers is simple yet surprisingly effective compared to the prior algorithm, which involves randomly partitioning the points into two groups and taking the intersection of a subset of points with a small diameter in one group with the points in the other group.  Our deterministic approach always succeeds as opposed to the previous randomized approach, which might require multiple repeats (and thus more time) to succeed. This significantly simplifies the analysis, allows us to improve the range of $\alpha$ and remove the required lower bound on the predicted cluster size.
>
> For k-medians, our work is the first to propose an algorithm in this setting that actively removes outliers in the predicted cluster, whereas the prior algorithm assumes that the label error rate is low enough that a random subset of the points is likely to contain no false points. We agree with the reviewer that each step of the algorithm is relatively standard; the surprising fact here is that the combination of these steps are sufficient to improve the range of $\alpha$ all the way to the interval $[0,0.5)$, as well as to get a near $1 + 7\alpha$ approximation for sufficiently small $\alpha$. Note that 0.5 is a natural barrier. When $\alpha < 0.5$, there is a natural 1-1 mapping between the true clusters and the given clusters. When $\alpha > 0.5$, it is possible that for each given cluster, there is no true cluster accounting for the majority of the points and thus no obvious correspondence exists.
>
> We view the natural and simple steps of the algorithm as a strength: while some theoretical algorithms involve intricate steps only to obtain certain theoretical guarantees, our algorithm is simple and scalable to large datasets. Note that there are a lot of algorithms only involving simple steps: both Ergun et al. (2021) and our algorithms for the same problem of k-medians are simple but they are arguably completely different.

---

### Decision · Program_Chairs · 2023-01-20

**Decision:**

Accept: poster

**Justification For Why Not Higher Score:**

The reviewers lack enthusiasm as far as the extent of the novelty of contributions, and mentioned that algorithmically the contributions are less exciting

**Justification For Why Not Lower Score:**

The paper has merits listed above that reviewers were unanimously somewhat positive on

**Metareview: Summary, Strengths And Weaknesses:**

This paper presents new algorithms for clustering in the learning-augmented setting. It provides theoretical guarantees when noisy advice is given from an 'oracle' as well as a CD implementation of an algorithm for learning. There are a number of new contributions that are relevant to the community, though reviewers were reserved in finding the novelty somewhat limited and the presentation quality needing improvement. The authors have responded well to the author comments, but the paper should be improved with a clear exposition and statement of novelty in the contributions, among other suggestions in the reviews.

**Note From Pc:**

if the above contains the word "oral" or "spotlight" please see: "oral" presentation means -> notable-top-5% and "spotlight" means -> notable-top-25%. As stated in our emails, we are disassociating presentation type from AC recommendations

**Summary Of Ac-Reviewer Meeting:**

The reviewers were completely unresponsive to several attempts to initiate discussion. As it is only 'borderline' in terms of the score but the sentiment was unanimous on the weak accept side, I have decided to vote for acceptance given that the authors provide a thoughtful response to reviews (which the reviewers refused to discuss).